# Cost-Effectiveness of Dengue Vaccination in Indonesia: Considering Integrated Programs with *Wolbachia*-Infected Mosquitos and Health Education

**DOI:** 10.3390/ijerph17124217

**Published:** 2020-06-12

**Authors:** Auliya A. Suwantika, Angga P. Kautsar, Woro Supadmi, Neily Zakiyah, Rizky Abdulah, Mohammad Ali, Maarten J. Postma

**Affiliations:** 1Department of Pharmacology and Clinical Pharmacy, Faculty of Pharmacy, Universitas Padjadjaran, Bandung 40132, Indonesia; neily.zakiyah@unpad.ac.id (N.Z.); r.abdulah@unpad.ac.id (R.A.); 2Center of Excellence in Higher Education for Pharmaceutical Care Innovation, Universitas Padjadjaran, Bandung 40132, Indonesia; m.j.postma@rug.nl; 3Center for Health Technology Assessment, Universitas Padjadjaran, Bandung 40132, Indonesia; 4Department of Pharmaceutical and Pharmacy Technology, Faculty of Pharmacy, Universitas Padjadjaran, Bandung 40132, Indonesia; angga.prawira@unpad.ac.id; 5Unit of Global Health, Department of Health Sciences, University Medical Center Groningen (UMCG), University of Groningen, 9713 AV Groningen, The Netherlands; 6Faculty of Pharmacy, Universitas Ahmad Dahlan, Yogyakarta 55164, Indonesia; wsupadmi@yahoo.com; 7Faculty of Educational Sciences, Universitas Pendidikan Indonesia, Bandung 40154, Indonesia; emaa.laith@upi.edu; 8Unit of Pharmaco-Therapy, Epidemiology & Economics (PTE2), Department of Pharmacy, University of Groningen, 9713 AV Groningen, The Netherlands; 9Department of Economics, Econometrics & Finance, Faculty of Economics & Business, Groningen, University of Groningen, 9747 AE Groningen, The Netherlands

**Keywords:** dengue fever (DF), dengue haemorrhagic fever (DHF), dengue shock syndrome (DSS), cost-effective, incremental cost-effectiveness ratios (ICER)

## Abstract

Despite the fact that morbidity and mortality rates due to dengue infection in Indonesia are relatively high, a dengue vaccination has not yet been introduced. Next to vaccination, *Wolbachia*-infected mosquitoes and health education have been considered to be potential interventions to prevent dengue infection in Indonesia. This study was aimed to analyse the cost-effectiveness of dengue vaccination in Indonesia whilst taking *Wolbachia* and health education programs into account. An age-structured decision tree model was developed to assess the cost-effectiveness. Approximately 4,701,100 children were followed-up in a 10-year time horizon within a 1-year analytical cycle. We compared three vaccination strategies: one focussing on vaccination only, another combining vaccination and a *Wolbachia* program, and a third scenario combining vaccination and health education. All scenarios were compared with a no-intervention strategy. The result showed that only vaccination would reduce dengue fever (DF), dengue haemorrhagic fever (DHF), and dengue shock syndrome (DSS) by 123,203; 97,140 and 283 cases, respectively. It would save treatment cost at $10.3 million and $6.2 million from the healthcare and payer perspectives, respectively. The combination of vaccination and a *Wolbachia* program would reduce DF, DHF and DSS by 292,488; 230,541; and 672 cases, respectively. It would also save treatment cost at $24.3 million and $14.6 million from the healthcare and payer perspectives, respectively. The combination of vaccination and health education would reduce DF, DHF, and DSS by 187,986; 148,220; and 432 cases, respectively. It would save treatment cost at $15.6 million and $9.4 million from the healthcare and payer perspectives, respectively. The incremental cost-effectiveness ratios (ICERs) from the healthcare perspective were estimated to be $9995, $4460, and $6399 per quality-adjusted life year (QALY) gained for the respective scenarios. ICERs from the payer perspective were slightly higher. It can be concluded that vaccination combined with a *Wolbachia* program was confirmed to be the most cost-effective intervention. Dengue infection rate, vaccine efficacy, cost of *Wolbachia* program, underreporting factor for hospitalization, vaccine price and mortality rate were considered to be the most influential parameters affecting the ICERs.

## 1. Introduction

As a major public-health concern in tropical and sub-tropical regions, dengue is the most rapidly spreading mosquito-borne viral disease in the world with a 30-fold increase in its infected cases over the last five decades [1,2]. Dengue is caused by the dengue virus (DENV) that has four different serotypes (DENV-1, DENV-2, DENV-3, and DENV-4). The circulation of several serotypes in a single region increases the probability of infection, the risk of epidemics and the severity of clinical manifestations of dengue in cases with second infections [3]. The manifestations of DENV infection range from dengue fever (DF), to the more severe forms, dengue haemorrhagic fever (DHF) and dengue shock syndrome (DSS) [4].

Approximately 50–100 million dengue infections occur annually and according to the World Health Organization (WHO), almost half of the world’s population lives in countries where dengue is endemic [2]. In particular, about 75% of the global population exposed to dengue are in the Asia-Pacific region [2]. As the most populous country in the Asia-Pacific region, Indonesia is consistently estimated to be among one of countries with the largest dengue burden in the world [5,6,7]. According to a recent study, the annual economic burden of dengue in Indonesia was estimated to be USD 381 million for hospitalized (USD 355 million) and ambulatory care cases (USD 26 million) [8].

Despite substantial investments, several existing prevention strategies have been proven to be insufficient in controlling dengue sustainably [9]. Nevertheless, dengue vaccination has been confirmed to be a cost-effective strategy to control dengue infection in a lot of endemic countries [10]. Specifically, it has been confirmed to be a highly cost-effective or even cost-saving intervention [11]. As one of the countries where dengue is endemic, up to now, dengue vaccination has not yet been included in the national immunization program in Indonesia. In general, with respect to dengue preventive strategies, a study conducted by Achee et al. in 2015 highlighted that there was no single intervention that would be sufficient to combat dengue infection in the world [12]. Therefore, an integrated evidence-based approach that is specifically targeted to local conditions will also be required for Indonesia [13].

A novel arbovirus vector control strategy by releasing *Wolbachia*-infected mosquitoes in Yogyakarta, one of provinces in Indonesia with a high incidence rate of dengue infection, has shown promising results on reducing dengue infection in that region [14,15]. *Wolbachia* is maternally inherited through cytoplasmic incompatibility that has evolved mechanisms to transmit itself directly or indirectly into host populations [16]. An added advantage of this strategy was reported in that *Wolbachia* could reduce replication of other arboviruses within the mosquito (e.g., chikungunya, yellow fever, and Zika viruses) and could potentially offer an adequate longer-term strategy [17,18,19]. In addition, community behaviour also has an important role to control dengue transmission in Indonesia. The community should have adequate knowledge, positive attitudes, and good practices in preventing dengue infection [20]. A study conducted by Arneliwati et al. in 2018 recommended that health workers in Indonesia should provide health education by using audio-visual media in the prevention of dengue fever, based on their findings that a significant improvement in the attitude and actions of families concerning the prevention of dengue infection was seen after the implementation of the health education intervention [21]. One important health education option concerns the use of audio-visual devices.

Despite the fact that the epidemiology and economic burden due to dengue infection in Indonesia are relatively high, a dengue vaccination has not yet been introduced. Additionally, *Wolbachia*-infected mosquitoes and health education using audio-visual devices have been considered as potential interventions to prevent dengue infection in Indonesia. This study aimed to analyse the cost-effectiveness of dengue vaccination in Indonesia combined with either a *Wolbachia* program or a health education program as an integrated approach to achieve dengue control.

## 2. Methods

An age-structured decision tree model was developed to investigate the cost-effectiveness of an integrated *Wolbachia* program or a health education program combined with a nationwide dengue vaccination in Indonesia (see Figure 1). Approximately 4,701,100 children, the number of Indonesian children in the age group of 9 years old, were considered as the target population for a nationwide dengue vaccination program that was in line with the recommended age for dengue vaccination in Indonesia [22]. The cohort was followed up over a 10-year time horizon with a 1-year analytical cycle by considering the highest seroprevalence in Indonesia in an age group that ranged between 9 and 18 years old [23]. Dengue infection (46.12 in 100,000 population) and mortality (0.83%) rates were estimated from rates over the last 10 years in Indonesia [24]. The results from two recent studies on the epidemiology of dengue virus infections in Indonesia were applied to estimate the probability of progression from DENV infection to DF (39.8%), and a continuation to DHF (59.8%) and to DSS (0.4%) [25,26]. Probabilities of outpatient and hospitalization cases (DF, DHF, and DSS) were based on a study by Nadjib et al. in 2015 [8]. Since the number of cases due to dengue infection in Indonesia are known to be underreported, adjustment factors for outpatient services and hospitalizations were applied at 45.90 and 7.65, respectively [8].

Applying the current situation as the baseline for comparison, we analysed three intervention scenarios: one scenario focused on vaccination only, another combined a *Wolbachia* program and vaccination, and a third scenario combined health education and vaccination. To estimate the effectiveness of the *Wolbachia* program, we applied the results from the latest study by O’Reilly et al., which estimated that a *Wolbachia* program would reduce the number of dengue-related outpatient, hospitalization, and fatal cases by 86% for all cases. It was assumed that individuals were susceptible and upon exposure would develop primary DENV infection [14]. In particular, to estimate the effectiveness of health education by using video information, we considered the results from a study on the effectiveness of education on enhancing families’ behaviours in preventing dengue in Indonesia. Audio-visual significantly contributed to changed behaviours in the aspects of information and persuasion by providing a stimulus to hearing and vision. This specific study showed changes in the levels of attitude (0.12; *p* = 0.007) and action (0.87; *p* = 0.000) of families in the prevention of dengue fever [21]. Considering the average of these numbers, we estimated that a health education using visual media would reduce dengue infection by 50% [21].

A vaccine efficacy of 44% was applied as estimated from a meta-analysis on 7 clinical trials using a random-effects model, with an estimated variation from 25 to 59% [27]. Currently, the registration of a recombinant, live-attenuated, tetravalent dengue vaccine (CYD-TDV) prescribes that it should be used within the indicated age range from the age of 9 years old. Applying a 3-dose vaccine that was given 6 months apart [28], we considered that the vaccination program could confer 10 years of immunity [29,30]. A vaccination coverage of 88% was estimated as derived from the average of complete immunization coverages in the last 10 years in Indonesia [24]. In the absence of available data on quality-adjusted life year (QALY) losses in Indonesia due to dengue infection, QALY losses in affected children were estimated from the international literature. In particular, the duration of illness was considered at 1 and 3.9 days for outpatient and inpatient cases, respectively, and disutility scores were used at 0.032 and 0.036, respectively [8,11]. For fatal cases, 1 QALY loss was assumed for each year after case fatality.

Cost analysis in this study was conducted from two perspectives: the healthcare (only direct medical costs) and payer perspectives (all costs covered by the Indonesian National Healthcare Insurance/BPJS Kesehatan). Healthcare costs of outpatient and hospitalization cases were derived from a study on the economic burden of dengue in Indonesia [8]. Payer costs of outpatient and hospitalization cases were derived from the tariffs of capitation and Indonesia case-based groups (INA-CBGs), respectively [31]. Costs of the *Wolbachia* program was estimated to be $3 per person in the targeted population, according to the global estimation to establish this program in a more dense population, such as Indonesia and Brazil [32]. In addition, cost of health education using visual media was estimated to be $0.02 per person in the targeted population by applying the proportion of information, education, and communication components in total healthcare costs during dengue outbreaks in Indonesia [33]. A vaccine price of $20 was applied from a study by Zeng et al., which focused on the cost-effectiveness of dengue vaccination in 10 endemic countries, including Indonesia [8]. Cost of vaccine administration ($3.42) and wastage (10%) were also derived from that study [8]. All cost items from different currencies and years were converted into 2018 US $ by using purchasing power parity (PPP) [34]. All costs were discounted with an annual rate of 3%. More detailed information on the parameters used in the model can be seen in Table 1.

The incremental cost-effectiveness ratios (ICERs) were evaluated by using the WHO’s criteria on cost-effectiveness of universal immunization according to the GDP per capita: (i) highly cost-effective (less than one GDP per capita); (ii) cost-effective (between 1 and 3 times GDP per capita); and (iii) cost-ineffective (more than 3 times GDP per capita) [36]. Univariate sensitivity analysis was performed to investigate the effects of different input parameters on cost and health outcomes. In addition, probabilistic sensitivity analysis (PSA) was performed by running 5000 Monte Carlo simulations. Budget impact analysis was performed by evaluating the affordability related to the required budget for vaccination (vaccination and treatment costs) from the healthcare perspective.

## 3. Results

Applying a cohort of 4,701,100 children [22], vaccination would only reduce DF, DHF, and DSS by 123,203; 97,140; and 283 cases, respectively. It would reduce DF by 114,028; 8943; and 232 for outpatient, hospitalization, and fatal cases, respectively. It would reduce DHF by 65,760; 31,032; and 349 for all cases, respectively. It would also reduce DSS by 0; 281; and 2 for all cases, respectively. Vaccination combined with a *Wolbachia* program would reduce DF, DHF, and DSS by 292,488; 230,541; and 672 cases, respectively. It would reduce DF by 270,731; 21,206; and 551 for outpatient, hospitalization and fatal cases, respectively. It would reduce DHF by 156,131; 73,582; and 828 for all cases, respectively. It also would reduce DSS by 0; 666; and 6 for all cases, respectively. Vaccination combined with health education would reduce DF, DHF, and DSS by 221,080; 174,313; and 508 cases, respectively. It would reduce DF by 204,616; 16,048; and 416 for outpatient, hospitalization, and fatal cases, respectively. It would reduce DHF by 118,002; 55,685; and 625 for all cases, respectively. It would also reduce DSS by 0; 504; and 4 for all cases, respectively (see Figure 2).

In the context of treatment cost, vaccination only would save treatment costs at $10.3 million and $6.2 million from the healthcare and payer perspectives, respectively. Vaccination combined with a *Wolbachia* program would save treatment cost at $24.3 million and $14.6 million from healthcare and payer perspectives, respectively. In addition, vaccination combined with health education would save treatment cost at $18.4 million and $11.1 million from healthcare and payer perspectives, respectively (see Figure 3). Furthermore, the ICERs from the healthcare perspective were estimated to be $9995; $4460; and $5374 per QALY gained in vaccination only, vaccination combined with a *Wolbachia* program, and vaccination combined with health education, respectively. From the payer perspective, the ICERs would be $10,174; $4639; and $5554 per QALY gained in respective scenarios (see Figure 4). Considering the GDP per capita in Indonesia of $3859 [35], the results confirmed that all scenarios would be cost-effective from both perspectives since the ICERs were between 1 and 3 times GDP per capita.

The effects of varying input parameters on the ICERs from the healthcare perspective are shown in a tornado chart. In vaccination only, sensitivity analysis showed that dengue infection rate, vaccine efficacy, underreporting factor for hospitalization, vaccine price, and case fatality rate were considered to be the most influential parameters affecting the ICERs (see Figure 5a). In vaccination combined with a *Wolbachia* program, sensitivity analysis showed that dengue infection rate, underreporting factor for hospitalization, cost of *Wolbachia* program, vaccine price, and case fatality rate were considered to be the most influential parameters affecting the ICERs (see Figure 5b). In vaccination combined with health education, dengue infection rate, underreporting factor for hospitalization, vaccine price, vaccine efficacy, and case fatality rate were considered to be the most significant parameters impacting the cost-effectiveness value (see Figure 5c).

Applying a threshold ICER of $4460 (ICER in vaccination combined with a *Wolbachia* program), the probability for the vaccination program from the healthcare perspective to be cost-effective would be 0%, 48.5%, and 0% in vaccination only, vaccination combined with a *Wolbachia* program, and vaccination combined with health education, respectively. Applying a threshold ICER of $11,577 (3× GDP per capita), the probability for the vaccination program from the healthcare perspective to be cost-effective would be 100% in all scenarios (see Figure 6).

The affordability related to the required budget of programs from the healthcare perspective are shown in cost-effectiveness affordability curves. Dengue vaccination with a vaccine price of $20 per dose would be implementable when the budget exceeds $254.48 million (see Figure 7a). In particular, dengue vaccination combined with a *Wolbachia* program would be implementable when the budget exceeds $268.37 million (see Figure 7b). Dengue vaccination combined with health education would be implementable when the budget exceeds $246.42 million (see Figure 7c). These required budgets would be approximately 84–92% of the routine immunization budget ($293 million) and 6% of the national healthcare budget ($4502 million).

## 4. Discussion

Dengue is a mosquito-borne viral disease with a large epidemiologic and economic burden in Indonesia [1]. Even though intense efforts have been conducted to control dengue infection, existing vector control remains ineffective [9]. Novel arbovirus vector control tools are needed and *Wolbachia*-infected mosquitoes are considered as an alternative approach to reduce dengue virus transmission significantly [37,38,39]. In Indonesia, a *Wolbachia* program has showed promising entomological results, as piloted in Yogyakarta [14,15]. Furthermore, community behaviour also plays an important role in dengue transmission. Previous studies mentioned that people’s adequate knowledge, positive attitudes, and good practices were associated with the output of dengue prevention in the community [40,41]. Despite the fact that various health promotions have been implemented by the government of Indonesia to increase community behaviour, negative behaviour related to dengue prevention remains exist, leading to increasing annual numbers of dengue cases. A recent study showed that the use of audio-visual media could significantly enhance the attitudes and actions of families in the prevention of dengue infection [21]. Potentially, vaccination is the most effective control strategy against dengue. As a public-health intervention, dengue vaccination has been proven to be cost-effective [42]. However, the introduction of this vaccine tends to be delayed in Indonesia due to the scarcity of local cost-effectiveness studies, inadequate health systems, and financial barriers [43]. Our study aimed to fill the gap concerning information around cost-effectiveness.

Three specific scenarios were compared in this study. Applying the GDP per capita cost-effectiveness threshold, all scenarios were considered to be cost-effective in the context of cost per QALY gained. This study is in line with other economic evaluation studies of new vaccination programs in Indonesia that concluded that these vaccination programs could be cost-effective if implemented [44,45,46]. This study is also similar to the result of previous studies that specifically investigated the cost-effectiveness of dengue vaccination in endemic countries, confirming that vaccination would be cost-effective in such settings [11,42]. Several factors tend to make dengue vaccination particularly favourable in an endemic country, such as high incidence of dengue, high vaccination impact, and high cost per case [11]. Comparing all scenarios, the result confirmed that combining vaccination with a *Wolbachia* program within an integrated approach was considered to be the most cost-effective intervention. A previous study investigated the potential cost-effectiveness of a *Wolbachia* program in Indonesia, which confirmed that *Wolbachia* released in high density urban areas is expected to be highly cost-effective and could potentially be a cost-saving intervention to prevent dengue infections [47]. In particular, regions with a strong public health infrastructure, fiscal capacity, and community support should be prioritized. Given that a *Wolbachia* program is also not predicted to fully eliminate dengue virus transmission in highly endemic settings, there is a need to understand how *Wolbachia* program interact with other interventions of dengue prevention and how the optimal package of interventions may change in different environments [14,47].

The results confirmed that the ICER from the healthcare perspective is lower than that from the payer perspective in all scenarios. Yet, there is no significant difference in the ICERs for both perspectives since treatment cost was not found to be the most influential parameter in the sensitivity analysis. The results of sensitivity analysis in our study shows that dengue infection rates, underreporting factors for hospitalization, cost of the *Wolbachia* program, vaccine price, vaccine efficacy, and case fatality rates were considered to be the most influential parameters affecting the cost-effectiveness of integrated dengue vaccination programs. Previous model analyses found the same parameters being most influential in sensitivity analyses [8,30,47,48,49,50].

Despite the fact that this study is not the first economic evaluation study on dengue vaccination in Indonesia, it has several major strengths. Compared to a previous study that analyzed the cost-effectiveness of dengue vaccination in Indonesia as one of ten endemic countries [11], several significant differences in the process of analyses were found. The main strength of this study is its use of country-specific data in the hypothetical model. Vaccine efficacy and utilities are the only parameters that were not derived from local data. Another strength is its application of two perspectives: the healthcare and payer perspectives. The healthcare perspective considers only direct medical costs, which is important for decision makers in the healthcare sector on making decisions. The payer perspective considers all cost covered by national healthcare insurance, which can be considered relevant in the context of the new healthcare system that Indonesia has implemented since 2014. Specifically, vaccination programs have not yet been included in the benefit package of the national healthcare insurance and our study provides an insight into the consequences of its potential inclusion. In addition, this study also analyzed the cost-effectiveness of dengue vaccination in Indonesia together with taking a *Wolbachia* program or a health education program into account. In line with the international consensus, the implementation of a combined intervention to combat dengue infection is more favorable rather than a single intervention, with the budget that comes with it, as shown in this study.

Nevertheless, this study has several limitations. The first and main limitation is its use of a static model instead of dynamic model. It is known that static models tend to over-estimate the ICERs due to their inability to incorporate the herd effect. However, if we took herd effect into account, there would be an even more favorable cost-effectiveness. The second limitation concerns the lack of country-specific data on vaccine efficacy. This data was applied from a meta-analysis by using the random-effects model [26]. Although in the evidence hierarchy, a well-designed meta-analysis is at the top of the pyramid [51], efficacy may be region-specific, as it is for some other vaccines like the rotavirus [52]. To deal with this limitation, we take this issue into account in sensitivity analyses.

This study provides information for stakeholders in Indonesia to develop the next comprehensive step to prevent dengue infection. To implement scenarios of vaccination only, vaccination combined with a *Wolbachia* program or vaccination combined with health education, the government of Indonesia would require a budget of $246-268 million, which is approximately 84–92% of its routine immunization budget ($293 million) and 6% of its national healthcare budget ($4502 million). As a country with limited healthcare and routine immunization budgets, this additional budgetary requirement would be very challenging. Since efforts to create new potential revenue remains limited, the most realistic approach to expand fiscal space in this situation is through efficiency gains in all healthcare programs. To conclude, vaccination combined with a *Wolbachia* program was confirmed to be the most cost-effective intervention. Nevertheless, It appears unrealistic to implement this without expanding fiscal space to finance immunization programs in Indonesia.

## 5. Conclusions

It can be concluded that vaccination combined with a *Wolbachia* program was confirmed to be the most cost-effective intervention, compared with vaccination only and vaccination combined with health education. Dengue infection rate, vaccine efficacy, cost of *Wolbachia* program, underreporting factors for hospitalization, vaccine price, and mortality rate were considered to be the most influential parameters affecting the ICERs.

## Figures and Tables

**Figure 1 ijerph-17-04217-f001:**
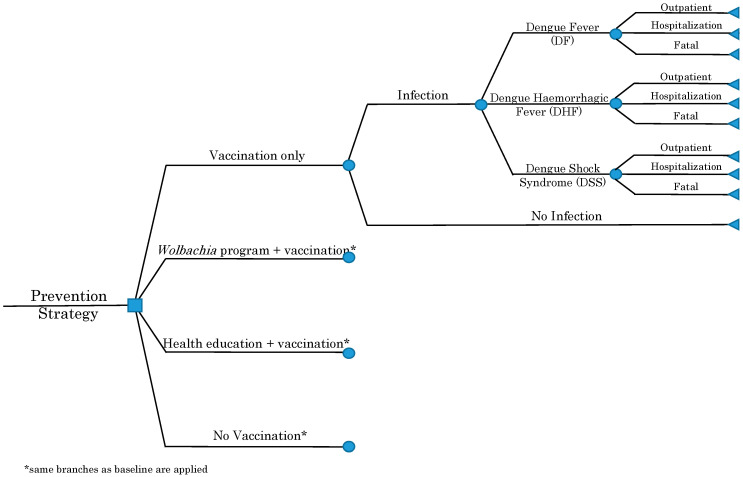
Decision tree model.

**Figure 2 ijerph-17-04217-f002:**
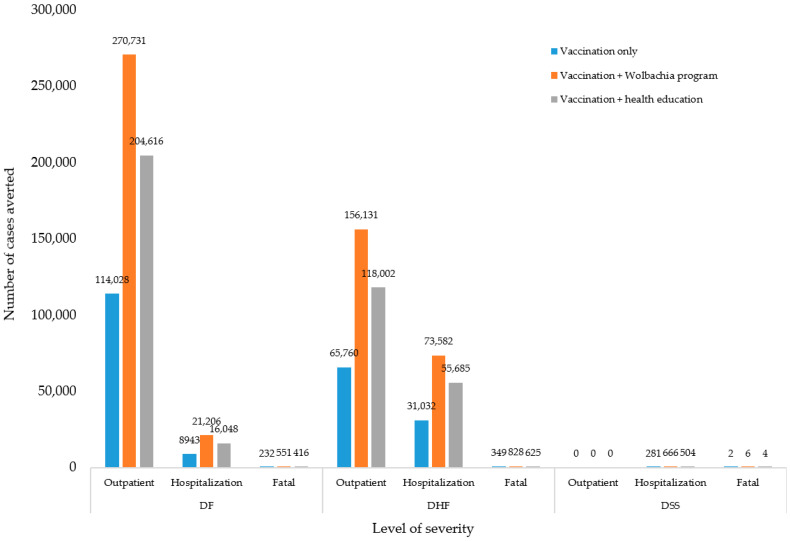
Number of cases averted.

**Figure 3 ijerph-17-04217-f003:**
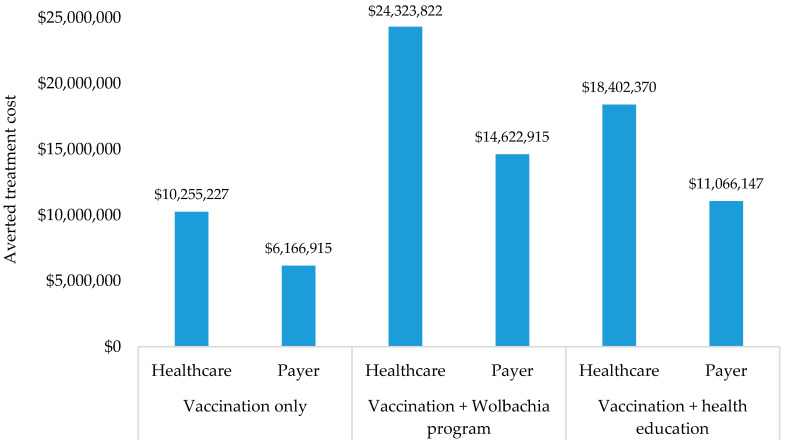
Averted treatment cost.

**Figure 4 ijerph-17-04217-f004:**
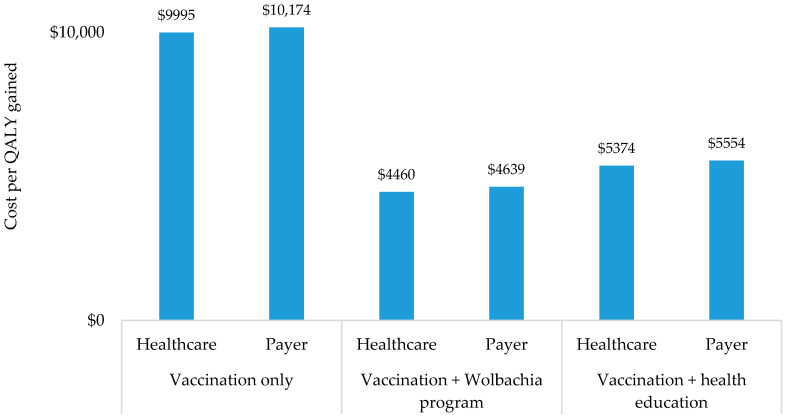
Incremental cost-effectiveness ratio.

**Figure 5 ijerph-17-04217-f005:**
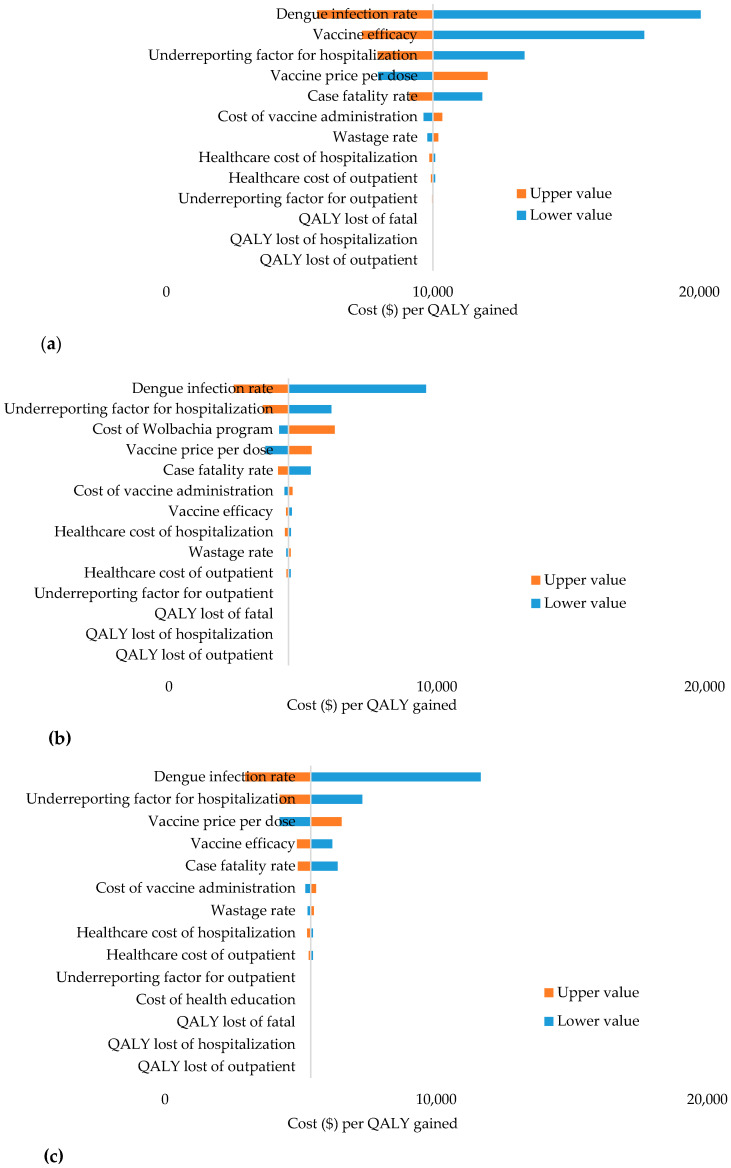
(**a**) Univariate sensitivity analysis (vaccination only); (**b**) Univariate sensitivity analysis (vaccination + *Wolbachia* program); (**c**) Univariate sensitivity analysis (vaccination + health education).

**Figure 6 ijerph-17-04217-f006:**
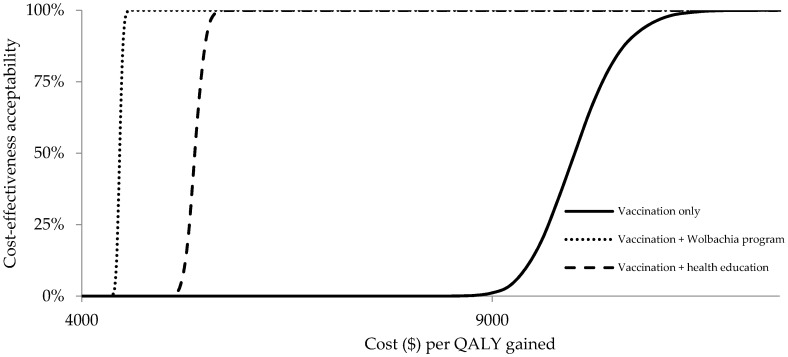
Cost-effectiveness acceptability curves from the healthcare perspective.

**Figure 7 ijerph-17-04217-f007:**
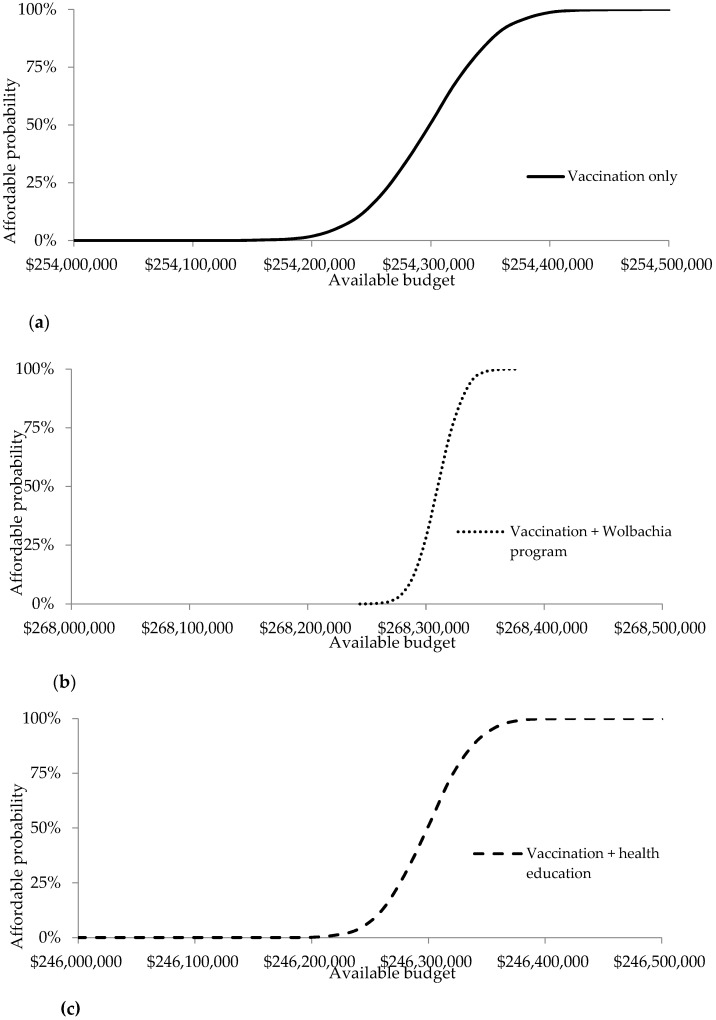
(**a**) Affordability curves from the healthcare perspective (vaccination only); (**b**) Affordability curves from the healthcare perspective (vaccination + *Wolbachia* program); (**c**) Affordability curves from the healthcare perspective (vaccination + health education).

**Table 1 ijerph-17-04217-t001:** Parameters used in the model.

Parameters	Value	Distribution	Reference
**Epidemiology**			
Dengue infection rate	0.05%	Dirichlet	[24]
Case fatality rate	0.83%	Dirichlet	[24]
Probability of DF	39.80%	Dirichlet	[25,26]
Probability of DHF	59.80%	Dirichlet	[25,26]
Probability of DSS	0.40%	Dirichlet	[25]
Probability of outpatient (DF)	68%	Dirichlet	[8]
Probability of hospitalization (DF)	32%	Dirichlet	[8]
Probability of outpatient (DHF)	26.10%	Dirichlet	[8]
Probability of hospitalization (DHF)	73.90%	Dirichlet	[8]
Probability of outpatient (DSS)	0%	Dirichlet	[8]
Probability of hospitalization (DSS)	100.00%	Dirichlet	[8]
Expansion factor for outpatient	45.90	Dirichlet	[8]
Expansion factor for hospitalization	7.65	Dirichlet	[8]
**Costs**			
Healthcare cost of outpatient	$19,22	Gamma	[8]
Healthcare cost of hospitalization	$297,79	Gamma	[8]
Payer cost of outpatient	$0,62	Gamma	[32]
Payer cost of hospitalization	$227,94	Gamma	[31]
Vaccine price per dose	$20,00	Gamma	[11]
Cost of vaccine administration	$3,42	Gamma	[11]
Cost of *Wolbachia* program	$3,00	Alternative scenario	[33]
Cost of health education	$0,02	Alternative scenario	[34]
Vaccine characteristics			
Vaccine efficacy	44.00%	Alternative scenario	[27]
Vaccine coverage	87.56%	Alternative scenario	[24]
Schedule (3-dose for >9 years old)	6-month interval		[30]
Wastage rate	10%	Alternative scenario	[11]
**Utilities**			
QALY loss of outpatient	0.00009	Beta	[8,11]
QALY loss of hospitalization	0.00038	Beta	[8,11]
QALY loss of fatal	1	Beta	[8,11]
**Others**			
Targeted population	4,701,100	Unvaried	[22]
Discount rate	3.00%	Unvaried	[35]
Time horizon	10 years	Unvaried	

QALY = quality-adjusted life year

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
