# Peer review of "Cost-Effectiveness of Dengue Vaccination in Indonesia: Considering Integrated Programs with Wolbachia-Infected Mosquitos and Health Education"

_ijerph, 2020, doi:10.3390/ijerph17124217_

Round 1

Reviewer 1 Report

Thank you for giving me the opportunity to review this very interesting paper.

The article is already very good in its current form, complete and understandable.

There are some minor improvements that could be made:

Introduction:

Line 80: I think it is necessary to explain more to the unfamiliar reader what Wolbachia infected mosquitoes are, as it is essential to understand why this technology/strategy is included in the authors' model.

Methods :

Line 102: We need more information about this cohort of 4'710'000 children, how it is recruited and who the children are.

Result :

We have to modify the figures 5 a-b-c, the histograms are superimposed on the text and we can't read the figure anymore.

Author Response

Introduction

Line 80: I think it is necessary to explain more to the unfamiliar reader what Wolbachia infected mosquitoes are, as it is essential to understand why this technology/strategy is included in the authors' model.

Response:

A novel arbovirus vector control strategy by releasing Wolbachia-infected mosquitoes in Yogyakarta, one of provinces in Indonesia with a high incidence rate of dengue infection, has shown promising results on reducing dengue infection in that region. Wolbachia is maternally inherited through cytoplasmic incompatibility that has evolved mechanisms to transmit itself directly or indirectly into host populations. An added advantage of this strategy was reported in thatWolbachia could reduce replication of other arboviruses within the mosquito (e.g. chikungunya, yellow fever and Zika viruses) and could potentially offer an adequate longer-term strategy. We have clarified this better now in the fourth paragraph of introduction section.

Methods

Line 102: We need more information about this cohort of 4,710,000 children, how it is recruited and who the children are.

Response:

Approximately 4,701,100 children, the number of Indonesian children in the age group of 9 years old, were considered as the target population for nationwide dengue vaccination program that was in line with recommended age for dengue vaccination in Indonesia. The cohort was followed-up over a 10-year time horizon with a 1-year analytical cycle by considering the highest seroprevalence in Indonesia in an age group of 9-18 years old. We have clarified this better now in the first paragraph of methods section.

Result

We have to modify the figures 5 a-b-c, the histograms are superimposed on the text and we can't read the figure anymore.

Response:

Figures have been modified accordingly.

Reviewer 2 Report

Well written manuscript, the methodology is fine but the presentation needs to improved!

  1. the conclusion is vaccination combined with a Wolbachia program was confirmed to be the most cost-effective intervention, compared with vaccination only and vaccination combined with health education. If so, I suggest that the figure of the sensitivity analysis between vaccination with Wolbachia program comparing to vaccination with health education to be presented in the text.
  2. Meanwhile, only the single variable sensitivity analysis with tornado plot were given, why not to present the PSA and CEAC?   

Author Response

Well written manuscript, the methodology is fine but the presentation needs to improved!

  1. The conclusion is vaccination combined with a Wolbachia program was confirmed to be the most cost-effective intervention, compared with vaccination only and vaccination combined with health education. If so, I suggest that the figure of the sensitivity analysis between vaccination with Wolbachia program comparing to vaccination with health education to be presented in the text.

Response:

We have added a figure on cost-effectiveness acceptability curves (CEACs) to compare the probability of combined interventions to be cost-effective in specific thresholds.

  1. Meanwhile, only the single variable sensitivity analysis with tornado plot were given, why not to present the PSA and CEAC? 

Response:

Univariate sensitivity analysis was performed to investigate the effects of different input parameters on cost and health outcomes. In addition, probabilistic sensitivity analysis (PSA) was performed by running 5,000 Monte Carlo simulations. Budget impact analysis was performed by evaluating the affordability related to the required budget for vaccination (vaccination and treatment costs) from the healthcare perspective in CEACs. We have added this information in the sections of methods and results.